# The Isolation, Screening, and Characterization of Polyhydroxyalkanoate-Producing Bacteria from Hypersaline Lakes in Kenya

**Martin N. Muigano** [1,*] **, Sylvester E. Anami** [1] **, Justus M. Onguso** [1] **and Godfrey M. Omare** [2]

[1] Institute for Biotechnology Research, Jomo Kenyatta University of Agriculture and Technology, Nairobi P.O. Box 62000-00200, Kenya
[2] Department of Physical and Biological Sciences, Bomet University College, Bomet P.O. Box 701-20400, Kenya
[*] Correspondence: muiganu@gmail.com

**Abstract:** Extremophilic microorganisms such as those that thrive in high-salt and high-alkaline environments are promising candidates for the recovery of useful biomaterials including polyhydroxyalkanoates (PHAs). PHAs are ideal alternatives to synthetic plastics because they are biodegradable, biocompatible, and environmentally friendly. This work was aimed at conducting a bioprospection of bacteria isolated from hypersaline-alkaliphilic lakes in Kenya for the potential production of PHAs. In the present study, 218 isolates were screened by Sudan Black B and Nile Red A staining. Of these isolates, 31 were positive for PHA production and were characterized using morphological, biochemical, and molecular methods. Through 16S rRNA sequencing, we found that the isolates belonged to the genera *Arthrobacter* spp., *Bacillus* spp., *Exiguobacterium* spp., *Halomonas* spp., *Paracoccus* spp., and *Rhodobaca* spp. Preliminary experiments revealed that *Bacillus* sp. JSM-1684023 isolated from Lake Magadi had the highest PHA accumulation ability, with an initial biomass-to-PHA conversion rate of 19.14% on a 2% glucose substrate. Under optimized fermentation conditions, MO22 had a maximum PHA concentration of 0.516 g/L from 1.99 g/L of cell dry weight and 25.9% PHA conversion, equivalent to a PHA yield of 0.02 g/g of biomass. The optimal PHA production media had an initial pH of 9.0, temperature of 35 °C, salinity of 3%, and an incubation period of 48 h with 2.5% sucrose and 0.1% peptone as carbon and nitrogen sources, respectively. This study suggests that bacteria isolated from hypersaline and alkaliphilic tropical lakes are promising candidates for the production of polyhydroxyalkanoates.

**Keywords:** isolation; screening; characterization; polyhydroxyalkanoates; PHA; Lake Magadi; Lake Simbi; *Bacillus* sp.



## 1. Introduction

Plastic pollution is one of the most pressing issues of the 21st century. Petroleum-based plastics are associated with environmental pollution due to their non-degradable nature and thus pose significant health hazards to humans and animal species [1]. The generation of plastic waste across the world more than doubled between 2000 and 2019 [2]. Recycling efforts have helped to ease the problem of single-use plastics but disposal in landfills still remains a major environmental challenge in many parts of the world [3]. As a result, there has been a renewed interest in the production of biodegradable polymers. Biodegradable polymers are readily disintegrated by life processes, and therefore, their utilization could minimize plastic pollution [4]. Thus, biodegradable polymers may play an important role in the creation of a circular economy with low carbon and high recycling efficiency [5].

Polyhydroxyalkanoates (PHAs) are among a group of notable biomaterials that have desirable qualities and the capacity to replace synthetic polymers. PHAs are polymeric substances produced by diverse microorganisms as energy and carbon storage materials when subjected to stressful conditions [6]. Generally, PHAs are produced when bacteria

cells are grown in a nutrient-deficient environment with an abundant carbon source [7–9]. PHAs possess essential features that make them suitable alternatives to hydrocarbon-based plastics, including high heat resistance and a melting point of up to 170 °C in addition to being biodegradable [10,11]. However, the production of polyhydroxyalkanoates is often limited by the high production costs compared with those of synthetic plastics [12–16]. To lower these costs, continuous research efforts have been directed towards the exploration of high-yielding strains and the optimization of fermentation conditions.

Extreme environments such as haloalkaliphilic lakes have diverse microbial communities that have the capacity for the production of polyhydroxyalkanoates. Halophiles have unique adaptation features for survival in hypersaline environments that make them attractive candidates for PHA production due to the reduced chances of microbial contamination in fermentation procedures [17,18]. The ability of halophiles to grow in environments with high salinity and alkalinity levels allows cost-effective production of polyhydroxyalkanoates in open, unsterile fermentation conditions [19]. As a result, halophiles have been explored as potential producers of polyhydroxyalkanoates. Some halophilic bacteria of the genus *Halomonas* spp. have emerged as suitable chassis microorganisms for the production of various polyhydroxyalkanoates [20]. Some alkalophilic bacteria may also produce polyhydroxyalkanoates as energy-storage compounds [21]. Therefore, halo-alkaliphilic microorganisms are promising sources of polyhydroxyalkanoates. Thus, the goal of this study was to prospect soda lakes in Kenya to identify bacteria that have the potential to produce polyhydroxyalkanoates.

## 2. Results

### 2.1. Isolation and Screening of PHA-Producing Bacteria

A total of 218 isolates were recovered from soil, sediment, and water samples from the haloalkaliphilic lakes. Of these, 31 showed bluish black coloration upon staining with a 3% ethanolic solution of Sudan Black B, thus suggesting that they could accumulate PHAs in their cells (Figure 1). Those that produced a negative test retained their original color. Microscopic examination showed the presence of black granules in all 31 isolates, 12 from Lake Simbi and 19 from Lake Magadi (Figure 2). All of the isolates showed orange fluorescence upon exposure to ultraviolet light after growth in media containing Nile red dye (Figure 3).

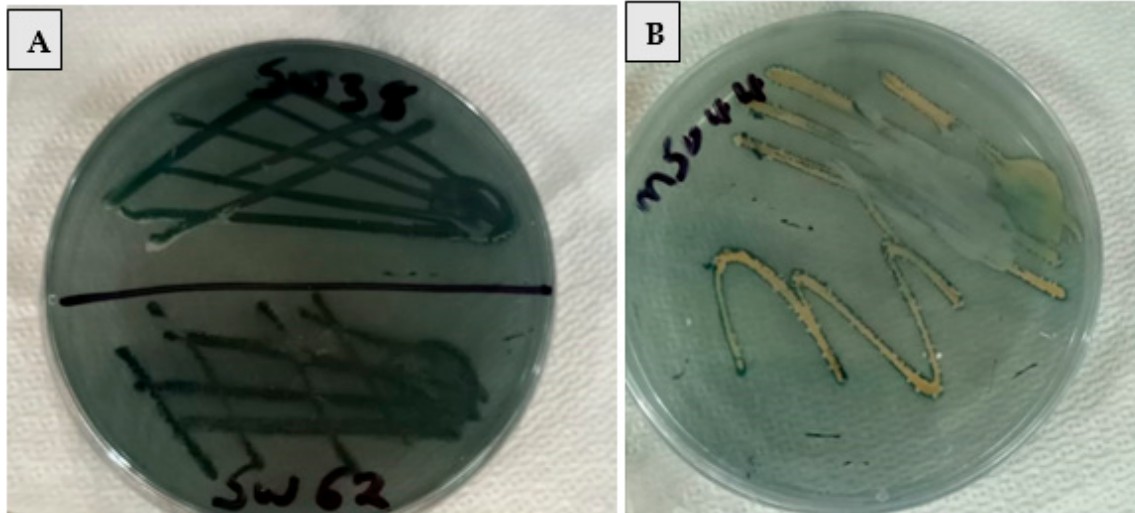

**Figure 1.** Bluish black coloration of PHA producers (**A**) and retention of original colony color by a non-PHA producer (**B**) after incubation with Sudan Black B dye.

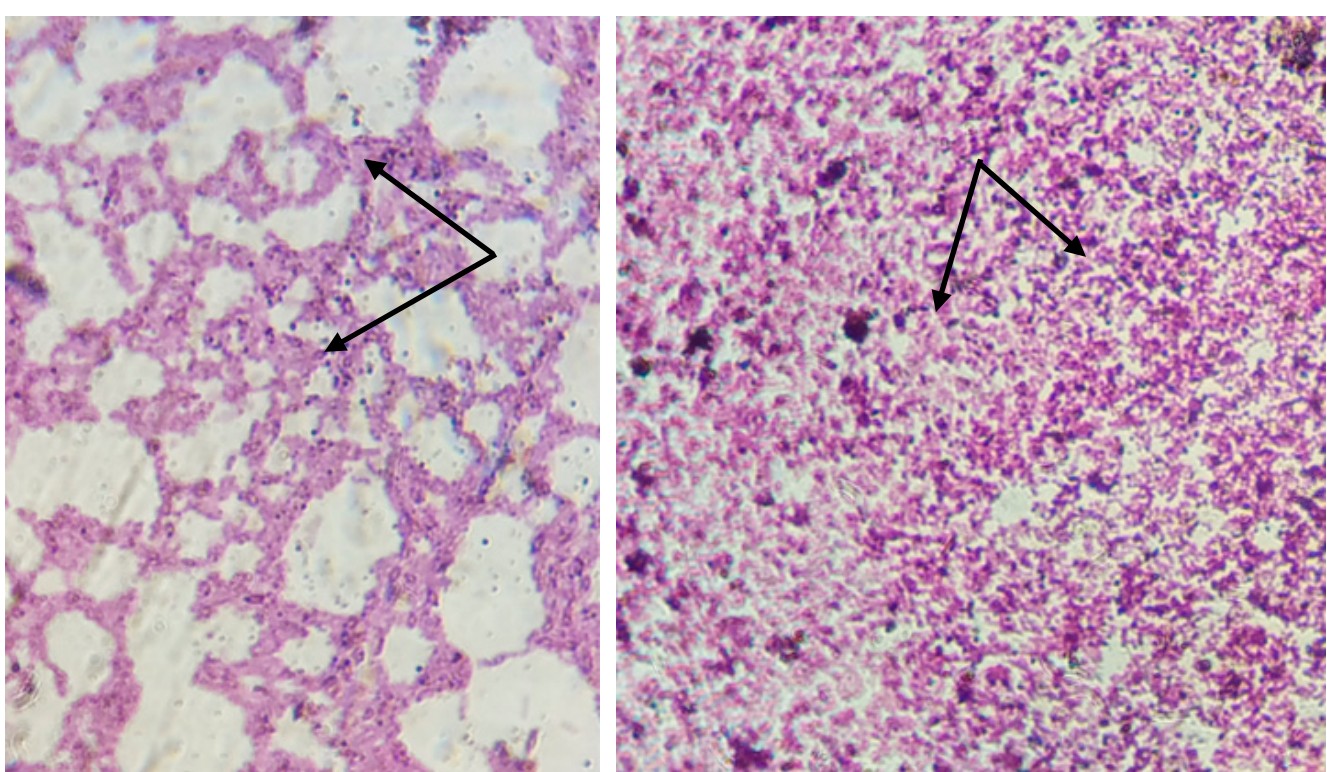

**Figure 2.** Presence of black granules on a PHA producer observed under a microscope. The dark-colored arrows point to the PHA granules on the microscope slide.

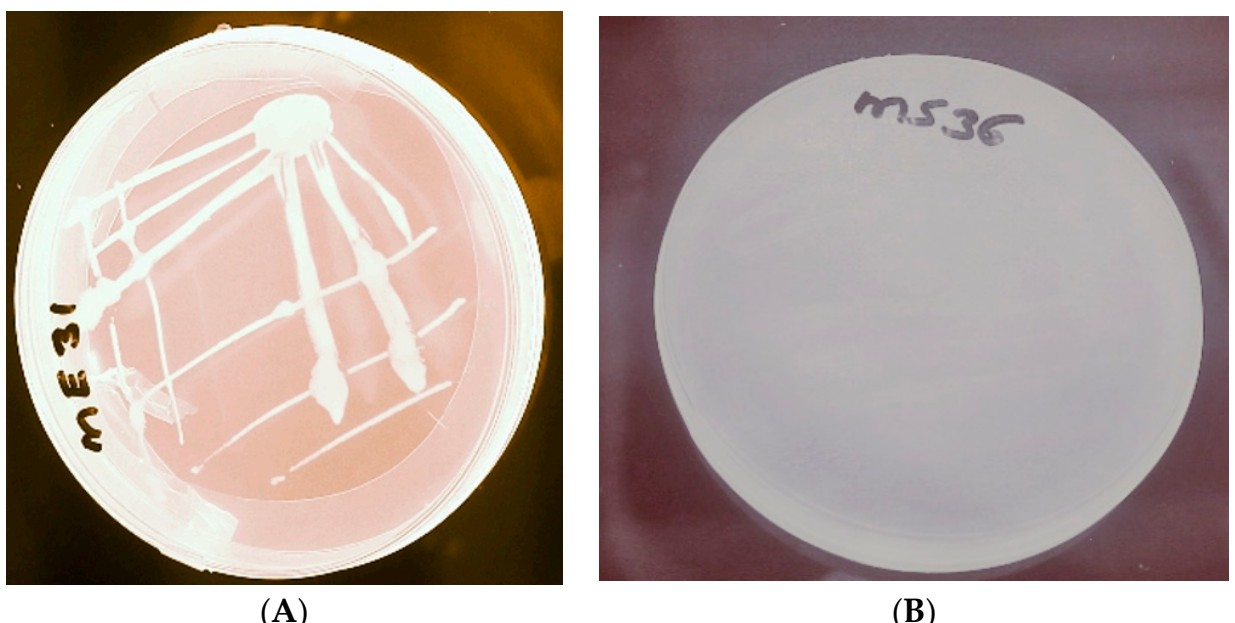

**(A)** **(B)**

**Figure 3.** Fluorescence observed on a PHA producer (**A**) and non-fluorescence on a non-PHA producer (**B**) grown on media containing Nile Red A.

*2.2. Characterization of PHA-Producing Bacteria*

2.2.1. Morphological and Biochemical Characterization

A total of 31 isolates were confirmed to have the ability to accumulate polyhydroxyalkanoates from Lakes Magadi and Simbi Nyaima. Morphological characterization of these isolates revealed the presence of a diverse group of microorganisms of various colony

morphologies. Gram staining results revealed that the majority of the isolates from Lake Magadi were gram-positive. Similarly, most PHA producers from Lake Simbi were gram-positive bacteria. The majority of the isolates were small rods with a circular cell structure and diverse pigmentation (Supplementary Materials).

The microorganisms had diverse biochemical features, as seen by the differential utilization of various substrates. In terms of carbohydrate catabolism, most isolates were found to metabolize glucose, sucrose, and fructose. However, other complex carbohydrates such as mannitol were not readily catabolized by some isolates. Table 1 below shows the morphological and biochemical characterizations of selected representative isolates: namely, SW32, SW36, MO22, and ME81. Isolate SW32 was identified as *Bacillus pumilus* through the sequencing of the 16S rRNA gene and was found to form gram-positive rods with yellowish pigmentation. The isolate was capable of catabolizing glucose, sucrose, fructose, galactose, lactose, maltose, mannose, and glycerol as sole carbon sources but unable to utilize starch or mannitol. On the other hand, isolate MO22, identified as *Bacillus* sp. strain JSM-1684023 through 16S rRNA gene sequencing, demonstrated yellow pigmentation with rod-shaped bacterial cells. MO22 had positive results for oxidase, catalase, Voges–Proskauer, and citrate utilization. It showed a capacity for the catabolism of common carbon sources but was unable to utilize starch or mannose as sole carbon sources. Isolate ME81, which was designated as *Bacillus* sp. Strain E-127, demonstrated cream pigmentation with rod-shaped cells and positive tests for gram staining, catalase, oxidase, Voges–Proskauer, and urease activity tests. Isolate SW36 was a gram-negative rod from Lake Simbi with an off-white pigmentation and an inability catabolize cellobiose. SW36, identified as *Halomonas alkalicola EXT*, was capable of glucose, sucrose, xylose, lactose, fructose, glycerol, galactose, and mannose catabolism. Table 1 shows the morphological and biochemical characterization of selected isolates.

**Table 1.** Morphological and biochemical characterization of selected PHA-producing bacteria.

| Characteristics | Isolates/Observations | | | |
| --- | --- | --- | --- | --- |
| | SW32 | SW36 | MO22 | ME81 |
| Colony shape | Circular | Circular | Circular | Circular |
| Color | Yellowish | Off white | Yellow | Cream |
| Cell shape | Rod | Rod | Rod | Rod |
| Gram reaction | + | − | + | + |
| Catalase | + | + | + | + |
| Oxidase | + | + | + | + |
| $H_2S$ production | − | − | − | − |
| Voges Proskauer | + | − | − | + |
| Urease activity | − | − | − | + |
| Indole production | − | − | − | − |
| Citrate utilization | + | − | + | + |
| Glucose | + | + | + | + |
| Sucrose | + | + | + | + |
| Xylose | + | − | + | + |
| Lactose | + | + | + | + |
| Starch | − | − | − | − |
| Maltose | + | − | + | + |
| Galactose | + | + | + | + |
| Fructose | + | + | + | + |
| Mannose | + | + | − | − |
| Cellobiose | + | − | + | + |
| Glycerol | + | + | + | + |
| Mannitol | − | − | + | − |

### 2.2.2. Molecular Characterization

Table 2 shows the species-level identities of PHA-producing bacteria isolated from Lakes Magadi and Simbi. Clonal analysis of the 16S rRNA gene on the DNAs of the PHA-producing bacteria showed that the isolates had diverse molecular characteristics. The clones of PHA-producing bacteria were close to six different genera: namely, *Arthrobacter* spp., *Bacillus* spp., *Exiguobacterium* spp., *Halomonas* spp., *Paracoccus* spp., and *Rhodobaca* spp. A majority of the isolates belonged to the *Bacillus* spp. genera, constituting 74.2% of all isolates.

A phylogenetic analysis revealed that the PHA-producing bacteria of the soda lakes in Kenya were predominantly of the *Bacillus* spp. genera (Figure 4). Bacteria of diverse genetic makeups were identified, as well as a few closely related microorganisms. However, greater diversity was observed in Lake Simbi compared with Lake Magadi. In Lake Simbi, representatives of four genera, including *Bacillus* spp., *Paracoccus* spp., *Halomonas* spp., *Arthrobacter* spp., and *Exiguobacterium* spp., were present in the sample. On the other hand, isolates from Lake Magadi included those of the genera *Bacillus* spp., *Paracoccus* spp., and *Exiguobacterium* spp.

**Table 2.** Species-level identities of PHA-producing bacteria isolated from Lake Simbi and Lake Magadi.

| Isolate Code | Organism | Query Cover | Percent Identity | Accession No |
|---|---|---|---|---|
| SO13 | *Bacillus* sp. 01105 | 99% | 98.52% | EU520307.1 |
| SO31 | *Bacillus subtilis* strain OTPB28 | 100% | 88.60% | KT265083.1 |
| SO75 | *Bacillus* sp. strain FA2-253 | 98% | 98.16% | KY476210.1 |
| SE42 | *Paracoccus* sp. TMN-21-1 | 100% | 96.76% | JX950033.1 |
| SE83 | *Exiguobacterium* sp. QZS4_8 | 99% | 97.51% | KX364032.1 |
| SE84 | *Arthrobacter* sp. strain C15 | 100% | 97.96% | MK182873.1 |
| SE89 | *Paracoccus* sp. TMN-21-1 | 99% | 97.62% | JX950033.1 |
| SE93 | *Bacillus* sp. ISO_06_Kulunda | 99% | 96.31% | EU676884.1 |
| SW32 | *Bacillus pumilus* strain 37 | 99% | 94.96% | MK327816.1 |
| SW36 | *Halomonas alkalicola* EXT | 100% | 97.49% | MK478810 |
| SW38 | *Bacillus* sp. strain MEB205 | 100% | 97.28% | MN809475.1 |
| SW62 | *Bacillus subtilis* strain QD9 | 98% | 98.52% | EF488088.1 |
| MO12 | *Bacillus* sp. KVD-DM52 | 97% | 88.90% | KJ872838.1 |
| MO15 | *Bacillus* sp. KVD-DM52 | 100% | 96.38% | KJ872838.1 |
| MO22 | *Bacillus* sp. strain JSM-1684023 | 96% | 96.80% | MG893133.1 |
| MO25 | *Rhodobaca* sp. strain ZN9W | 99% | 96.89% | MH463983.1 |
| ME31 | *Bacillus safensis* strain Ter61 | 95% | 97.41% | MW672512.1 |
| ME32 | *Bacillus* sp. strain CHA410 | 96% | 87.04% | MT355498.1 |
| ME33 | *Bacillus pumilus* strain 37 | 99% | 98.00% | MK327816.1 |
| ME42 | *Rhodobaca* sp. strain HJB301 | 100% | 97.47% | MT892652.1 |
| ME51 | *Bacillus* sp. E-127 | 95% | 95.44% | FJ764772.1 |
| ME54 | *Bacillus* sp. strain JSM 1684086 | 98% | 97.47% | MG893132.1 |
| ME81 | *Bacillus* sp. E-127 | 100% | 96.43% | FJ764772.1 |
| ME82 | *Bacillus* sp. ISO_06_Kulunda | 97% | 96.35% | EU676884.1 |
| ME88 | *Bacillus* sp. ISO_09_Wadi-Natrun | 95% | 96.85% | EU676885.1 |
| MS42 | *Bacillus* sp. ISO_06_Kulunda | 99% | 98.63% | EU676884.1 |
| MS44 | *Exiguobacterium aurantiacum* GBRS02 | 99% | 99.00% | MT373550.1 |
| MS52 | *Bacillus* sp. ISO_06_Kulunda | 99% | 98.63% | EU676884.1 |
| MW27 | *Bacillus* sp. A-09 | 76% | 76.47% | AY347311.1 |
| MW52 | *Bacillus* sp. BAB-1831 | 91% | 99.00% | KF771913.1 |
| MW54 | *Bacillus* sp. (in: firmicutes) | 99% | 97.49% | KX816446.1 |

### 2.3. Production of Polyhydroxyalkanoates by Bacteria Isolated from Hypersaline Lakes in Kenya

Bacteria isolates from all six genera, including *Arthrobacter* sp., *Bacillus* sp., *Exiguobacterium* sp., *Halomonas* sp., *Paracoccus* sp., and *Rhdodobaca* sp., produced polyhydroxyalkanoates of varying amounts in preliminary experiments. The PHAs were produced during the exponential

phase of growth in the PHA production medium after 72 h of incubation. *Halomonas alkalicola* Ext isolated from Lake Simbi produced 0.397 g/L of PHAs from 2.412 g of biomass, translating to a PHA content of 16.46%. *Bacillus* sp. JSM-1684023 from Lake Magadi had the highest PHA content of 19.14%, accounting for a 0.29 g/L PHA accumulation from 1.515 g/L of biomass. Other isolates with high PHA accumulation levels included strains SW32, SW38, and SW62 from Lake Simbi, which produced 0.235 g/L, 0.189 g/L, and 0.188 g/L, respectively (Table 3). From Lake Magadi, isolates ME33 and ME82 produced high PHA quantities of 0.215 g/L and 0.196 g/L, respectively. Strain SW32 (*Bacillus pumilus* strain 37) had a high PHA content of 13.26%. Similarly, strain MS42, identified as *Bacillus* sp. ISO_06_Kulunda, demonstrated a high PHA content of 13.16%. Isolates ME32 (*Bacillus* sp. strain CHA410), MO25 (*Rhodobaca* sp. strain ZN9W), MW54 (firmicutes *Bacillus* sp.), SE83 (*Exiguobacterium* sp. QZS4_8), and SO31 accumulated PHAs at concentrations below 5.0%. The PHA yields were below 0.01 g/g for most isolates upon preliminary screening, which were significantly lower than the maximum theoretical yield of 0.38 g/g for glucose substrates [22]. However, isolate SW36 had a relatively high yield of 0.02 g/g, while MO22 and MW27 yielded 0.015 g/g each.

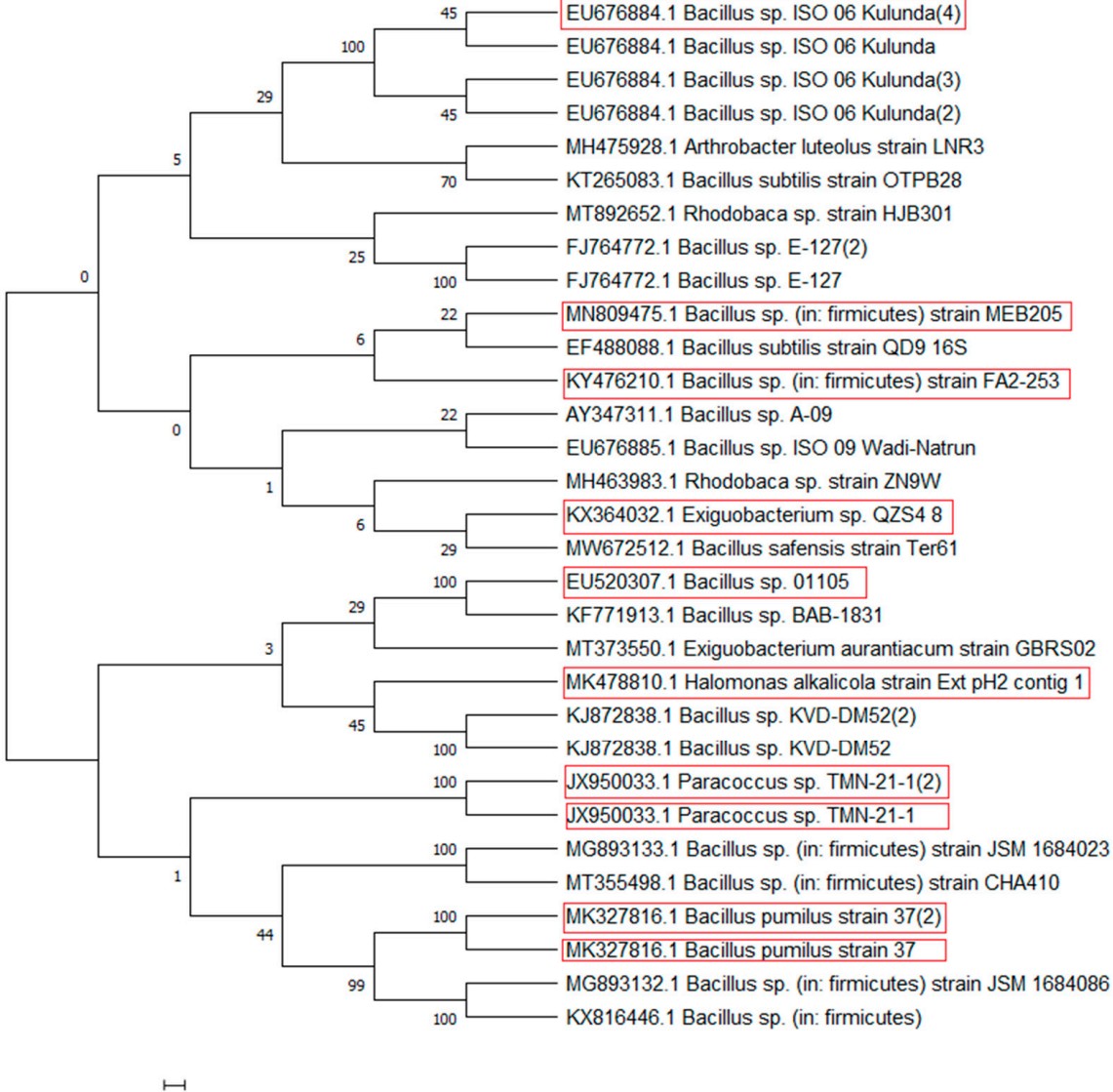

**Figure 4.** A phylogenetic tree of PHA-producing bacteria isolated from Lakes Magadi and Simbi. The isolates with red-colored frames were isolated from Lake Simbi, and the rest were from Lake Magadi.

### 2.4. Bacteria Cell Growth Dynamics

Cell growth was assessed for strains MO22 and SW36 to determine the optimal physiochemical factors. The effects of salinity, pH, and temperature were assessed on the optical density of cell cultures.

The growth dynamics of isolates MO22 and SW36 varied at different sodium chloride concentrations are shown in Figure 5. Optimal growth for isolates MO22 and SW36 was achieved at 0.5% and 3%, respectively. Both isolates showed high salt tolerance, with substantial growth at 10%. However, they were unable to grow at salt concentrations exceeding 20%. Overall, both isolates recorded the highest O.D. at lower salt concentrations. MO22 was found to have a huge pH range of between 6.0 and 11.0. However, optimal growth was observed at a pH of 10.0 for this isolate (Figure 5). On the other hand, SW36 grew optimally at a pH of 8.0, as shown in the figure below. With respect to temperature, both strains grew optimally at 30 °C and 35 °C. Both strains reached a stationary growth phase after 72 h of incubation. Figure 5 below shows the O.D. measurements after 48 h of incubation at different salinity levels, pH range, and temperatures. Growth was also traced at different time intervals.

**Table 3.** Preliminary production of polyhydroxyalkanoates by bacteria isolated from Lakes Simbi and Magadi. DCW and PHA concentrations represent means of triplicates. DCW = dry cell weight; PHA content = % of DCW. Isolates designated S and M were isolated from Lake Simbi and Lake Magadi, respectively.

| Isolate | Organism | DWC (g/L) | PHA Concentration (g/L) | PHA Content | PHA Yield (g/g) |
|---------|----------|-----------|-------------------------|-------------|-----------------|
| ME31 | *Bacillus safensis* strain Ter61 | 2.881 | 0.151 | 5.24% | 0.008 |
| ME32 | *Bacillus* sp. strain CHA410 | 2.117 | 0.090 | 4.25% | 0.005 |
| ME33 | *Bacillus pumilus* strain 37 | 1.890 | 0.215 | 11.38% | 0.011 |
| ME42 | *Rhodobaca* sp. strain HJB301 | 1.984 | 0.150 | 7.56% | 0.008 |
| ME51 | *Bacillus* sp. E-127 | 1.678 | 0.110 | 6.56% | 0.006 |
| ME54 | *Bacillus* sp. strain JSM 1684086 | 0.935 | 0.085 | 9.09% | 0.004 |
| ME81 | *Bacillus* sp. E-127 | 1.718 | 0.150 | 8.73% | 0.008 |
| ME82 | *Bacillus* sp. ISO_06_Kulunda | 1.610 | 0.196 | 12.17% | 0.010 |
| ME88 | Bacillus sp. ISO_09_Wadi-Natrun | 0.877 | 0.080 | 9.12% | 0.004 |
| MO12 | *Bacillus* sp. KVD-DM52 | 1.025 | 0.090 | 8.78% | 0.005 |
| MO15 | *Bacillus* sp. KVD-DM52 | 1.043 | 0.078 | 7.48% | 0.004 |
| MO22 | *Bacillus* sp. strain JSM-1684023 | 1.515 | 0.290 | 19.14% | 0.015 |
| MO25 | *Rhodobaca* sp. strain ZN9W | 2.619 | 0.108 | 4.12% | 0.005 |
| MS42 | *Bacillus* sp. ISO_06_Kulunda | 1.292 | 0.170 | 13.16% | 0.009 |
| MS44 | *Exiguobacterium aurantiacum* GBRS02 | 1.888 | 0.144 | 7.63% | 0.007 |
| MS52 | *Bacillus* sp. ISO_06_Kulunda | 1.305 | 0.095 | 7.28% | 0.005 |
| MW27 | *Bacillus* sp. A-09 | 3.506 | 0.275 | 7.84% | 0.015 |
| MW52 | *Bacillus* sp. BAB-1831 | 0.877 | 0.055 | 6.27% | 0.003 |
| MW54 | *Bacillus* sp. (in: firmicutes) | 1.458 | 0.064 | 4.38% | 0.003 |
| SE42 | *Paracoccus* sp. TMN-21-1 | 1.130 | 0.079 | 6.99% | 0.004 |
| SE83 | *Exiguobacterium* sp. QZS4_8 | 1.411 | 0.062 | 4.39% | 0.003 |
| SE84 | *Arthrobacter* sp. strain C15 | 0.689 | 0.052 | 7.55% | 0.003 |
| SE89 | *Paracoccus* sp. TMN-21-1 | 1.082 | 0.085 | 7.86% | 0.004 |
| SE93 | *Bacillus* sp. ISO_06_Kulunda | 1.360 | 0.151 | 11.10% | 0.008 |
| SO13 | *Bacillus* sp. *01105* | 1.194 | 0.078 | 6.53% | 0.004 |
| SO31 | *Bacillus subtilis* strain OTPB28 | 1.401 | 0.052 | 3.71% | 0.003 |
| SO75 | *Bacillus* sp. strain FA2-253 | 1.232 | 0.082 | 6.66% | 0.004 |
| SW32 | *Bacillus pumilus* strain 37 | 1.772 | 0.235 | 13.26% | 0.012 |
| SW36 | *Halomonas alkalicola* EXT | 2.412 | 0.397 | 16.46% | 0.020 |
| SW38 | *Bacillus* sp. strain MEB205 | 1.942 | 0.189 | 9.73% | 0.009 |
| SW62 | *Bacillus subtilis* strain QD9 | 3.520 | 0.188 | 5.34% | 0.009 |

### 2.5. Production of Polyhydroxyalkanoates on Different Carbon and Nitrogen Sources

The optimization of PHA production in this study was conducted on strain MO22. Figure 6 shows the PHA production on various carbon and nitrogen sources under optimized growth conditions. Optimal pH, temperatures, salinity levels, and incubation times were initially assessed on media containing 2% glucose and 0.2% ammonium sulfate as sources of carbon and nitrogen, respectively. Strain MO22 accumulated the highest PHA amount of 0.32 g/L at a pH of 9.0, while an optimal temperature of 35 °C produced 0.37 g/L of PHAs. The isolate demonstrated an optimal PHA production of 0.392 g/L at a salinity of 3%. An incubation time of 48 h produced the maximum PHA accumulation of 0.385 g/L. Thus, the ideal physiochemical conditions were identified as a pH of 9.0, a temperature of 35 °C, a 3% salinity, and an incubation time of 48 h for optimal PHA accumulation by strain MO22.

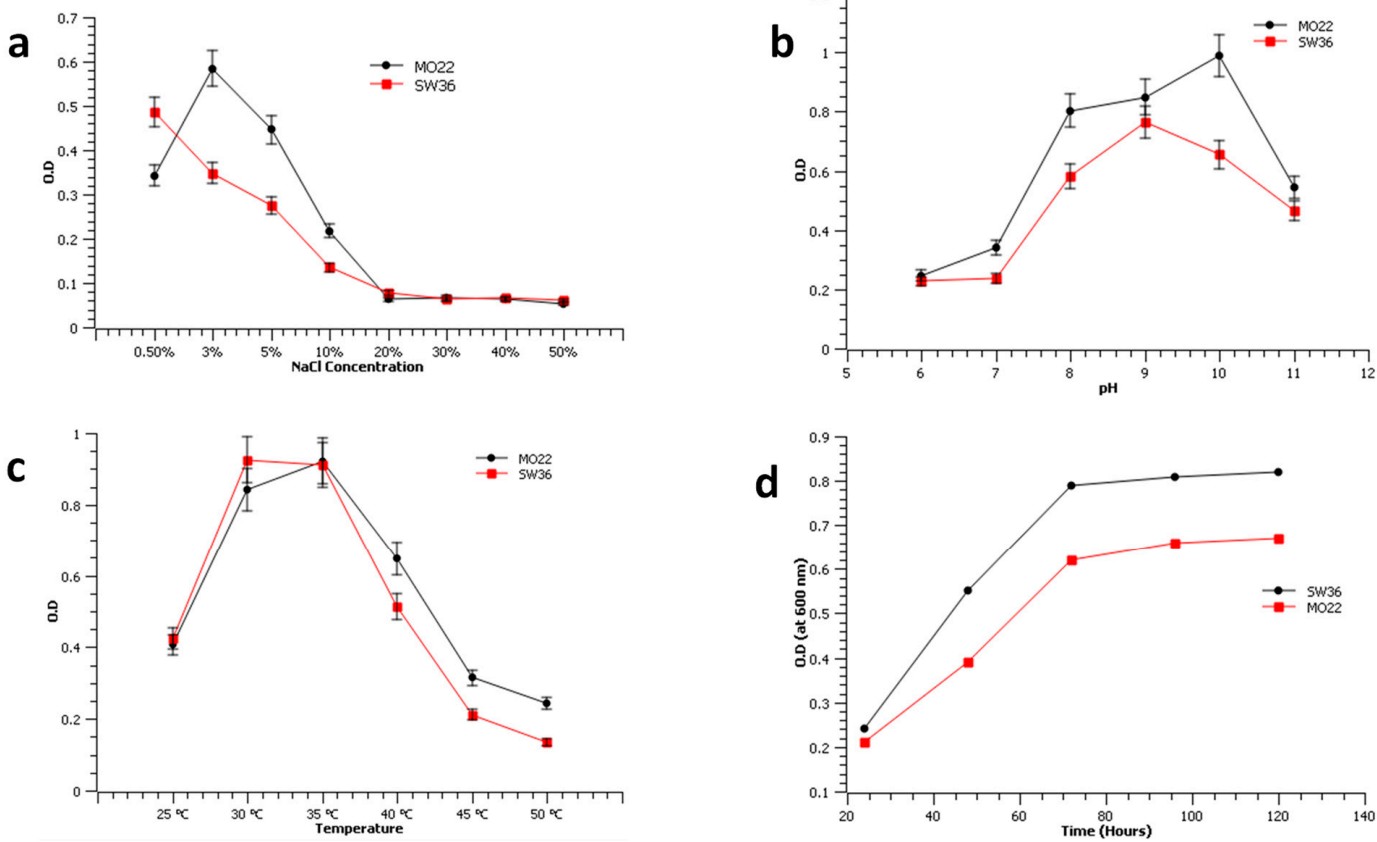

**Figure 5.** Cell growth dynamics at different salinity levels (**a**), pH (**b**), temperatures (**c**), and incubation times (**d**). Values represent means of triplicates.

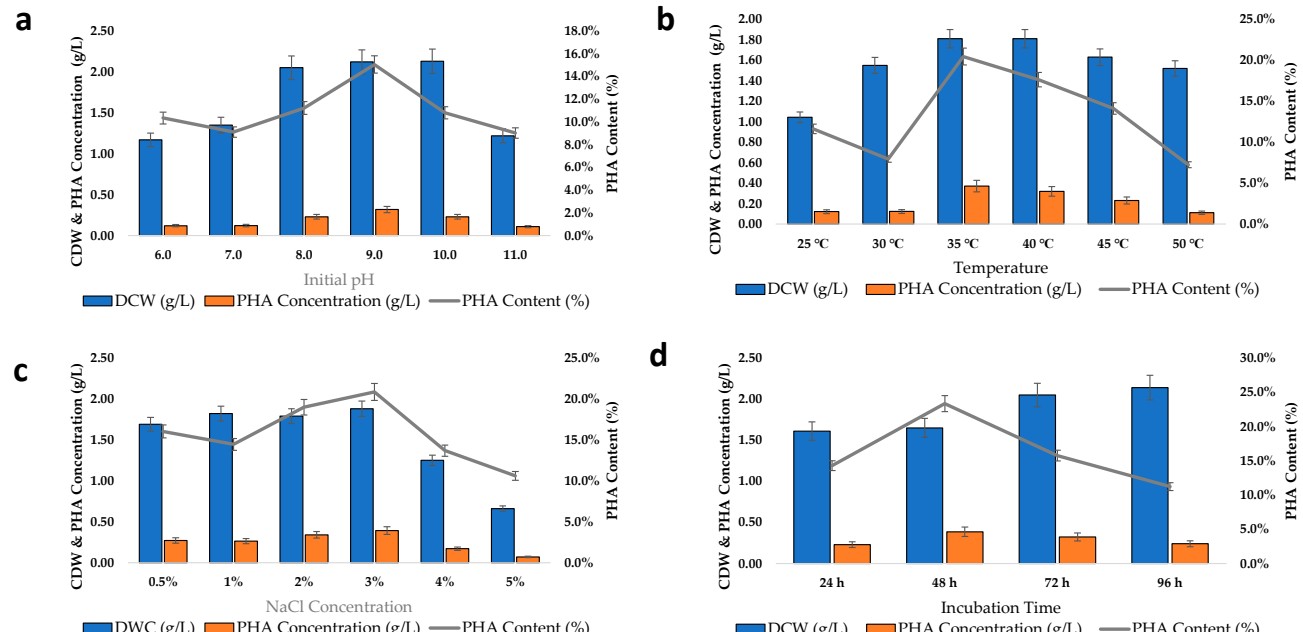

**Figure 6.** Optimized production of polyhydroxyalkanoates by MO22 on different initial pH levels (**a**), temperatures (**b**), salinity levels (**c**), and incubation times (**d**). Values show the means of measurements in triplicates, and error bars represent the standard deviations from the means.

Table 4 shows the production of PHAs by strain MO22 on different carbon sources. Production was conducted at an initial pH of 9.0, with a 3% NaCl concentration, and a temperature of 35 °C for 48 h. The results show that MO22 accumulated a maximum PHA concentration of 0.44 g/L when sucrose was used as a sole carbon source. This reflected a conversion rate of 21.9% from 2.01 g/L of biomass. A PHA yield of 0.02 g/g was achieved on 2% sucrose. Similarly, a maximum yield of 0.02 g/g was attained on glucose, fructose, and mannitol substrates. These values correspond to 4% of the maximum theoretical yield of 0.50 g/g for sucrose substrate [23] in the classical PHA formation pathway. This suggests that much of the carbon source went to bacteria growth rather than PHA production.

**Table 4.** Production of polyhydroxyalkanoates by strain MO22 on different carbon sources. DCW and PHA concentrations represent means of triplicates. * Maximum theoretical yield.

| Carbon Source | DCW (g/L) | PHA Concentration (g/L) | PHA Content (%) | PHA Yield (g/g) |
|---|---|---|---|---|
| Glucose | 2.07 | 0.39 | 18.89% | 0.02 |
| Fructose | 2.14 | 0.37 | 17.43% | 0.02 |
| Sucrose | 2.01 | 0.44 | 21.89% | 0.02 |
| Xylose | 1.42 | 0.21 | 15.00% | 0.01 |
| Galactose | 1.43 | 0.23 | 16.08% | 0.01 |
| Glycerol | 2.12 | 0.22 | 10.24% | 0.01 |
| Mannitol | 2.22 | 0.38 | 17.12% | 0.02 |
| * Sucrose [23] | - | - | - | * 0.50 |

Figure 7 shows the effects of different carbon and nitrogen sources as well as their concentrations on PHA production by strain MO22. While sucrose demonstrated the highest absolute PHA accumulation, equally high production levels were observed on glucose, fructose, and mannitol. Maximum PHA accumulation was achieved with a 2.5% sucrose concentration. Production reduced significantly with further increases in carbon source concentration. Moreover, strain MO22 showed a preference for peptone as a nitrogen source for PHA production, with a maximum PHA production of 0.514 g/L from 2.23 g/L of biomass, accounting for a 23.05% conversion. An evaluation of the effects of peptone

concentration on PHA accumulation revealed a maximum concentration of 0.516 g/L, and a PHA content of 25.9% was achieved on 0.1% peptone and 2.5% sucrose. This reflects a PHA yield of 0.021 g/g of biomass. However, no statistically significant differences were observed in PHA accumulation with 0.1% and 0.2% peptone concentrations.

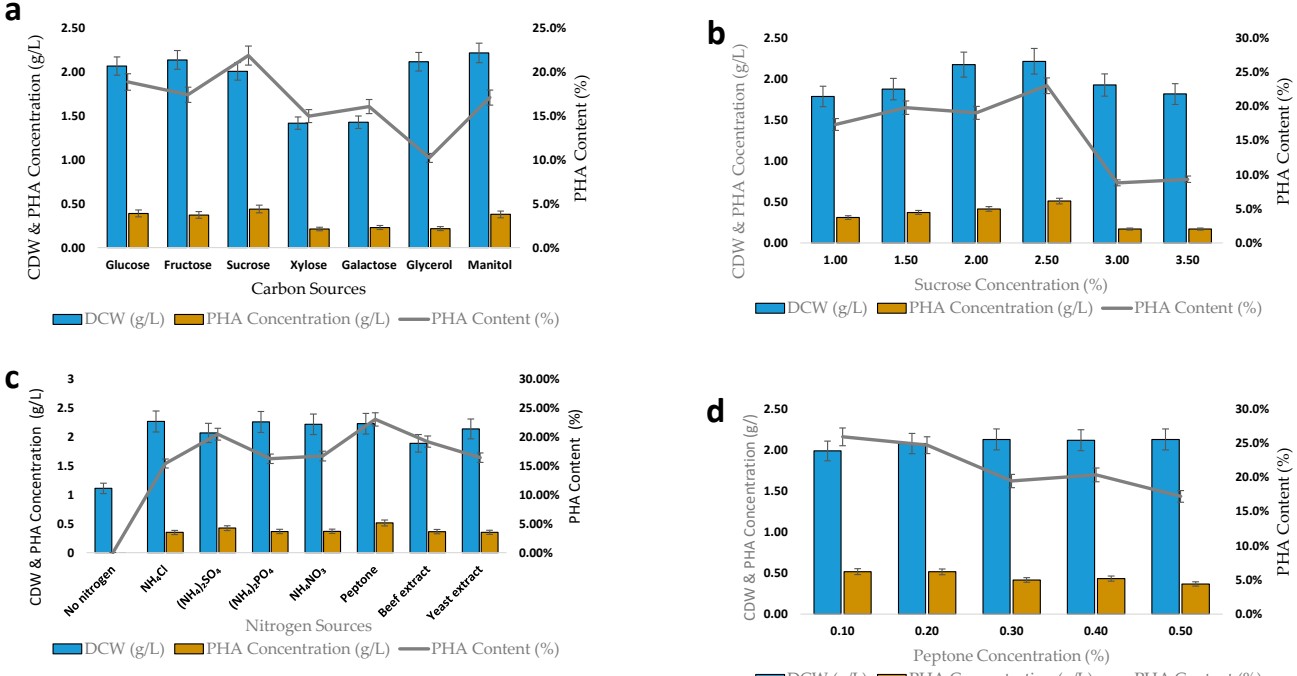

**Figure 7.** Effects of carbon sources (**a**), sucrose concentration (**b**), nitrogen sources (**c**), and concentration of peptone on the production (**d**) of polyhydroxyalkanoates by MO22. Values show the means of measurements in triplicates, and error bars represent the standard deviations from the means.

## 3. Discussion

Extremophiles have attracted significant attention due to their potential use in the production of industrially useful metabolites, including biodegradable polymers [24]. Hence, this study sought to evaluate the potential for the production of polyhydroxyalkanoates from two hypersaline lakes in Kenya: namely, Lake Magadi and Lake Simbi. The biosynthesis of polyhydroxyalkanoates by halotolerant bacteria has the advantage of minimizing contamination due to the extreme salt concentrations and pH levels, which reduce the chances of unwanted microbes growing in fermentation cultures. As such, the costs of PHA production may be minimized by eliminating the need for energy-intensive sterilization. Therefore, our study involved a bioprospection of potential PHA producers from hypersaline lakes in Kenya. After screening more than 200 isolates from the two lakes, 31 were found to be potential PHA producers and were characterized using morphological, biochemical, and molecular methods. The findings reveal that both lakes have diverse microbes of different morphological and biochemical features.

In our study, 31 PHA-producing bacteria isolates were identified as belonging to six different genera, including *Arthrobacter* spp., *Bacillus* spp., *Exiguobacterium* spp., *Halomonas* spp., *Paracoccus* spp., and *Rhdodobaca* spp. Most of the isolates belonged to the *Bacillus* group. This observation is consistent with prior findings that have demonstrated a predominance of the genus *Bacillus* in the synthesis of polyhydroxyalkanoates. According to Mizuno et al. [25], bacteria of the genus *Bacillus* produce class IV PHA synthase, an enzyme that is critical in the biosynthesis of polyhydroxyalkanoates. Other genera have also been shown to produce various types of polyhydroxyalkanoates [26,27]. For example, Das et al. have reported the recovery of poly(3-hydroxybutyrate) from *Exiguobacterium acetylicum* BNL 103 [28]. Bacteria of the genera *Alcaligenes*, *Burkholderia*, *Ralstonia*, *Pseudomonas*, *Paracoccus*,

and *Rhodobacter* have also been reported to accumulate various types of polyhydroxyalkanoates [29]. Members of the genus *Halomonas* have also shown potential for the production of polyhydroxyalkanoates [30–35]. Luo et al. reported a yield of up to 5.9 g from 100 g of bamboo biomass from *Halomonas alkalicola* [33]. A PHA concentration of 88.12% from 37.9 g/L of cell dry weight has been achieved in upscaled fermentation conditions by *Halomonas venusta* KT832796 [18]. Thus, bacteria of *Bacillus* and *Halomonas* genera are high-potential accumulators of polyhydroxyalkanoates.

Submerged fermentation experiments revealed that isolates MO22 and SW36 had strong potential for the production of polyhydroxyalkanoates. The newly isolated bacterium *Bacillus* sp. JSM-1684023 was identified as an attractive PHA producer based on preliminary experiments that showed the highest biomass-to-polymer conversion rate of 19.14% and a PHA accumulation of 0.29 g/L on 2% glucose. Under optimized conditions, strain MO22 accumulated a maximum concentration of 0.516 g/L from 1.99 g/L of biomass, with a 25.9% biomass to PHA conversion, equivalent to a PHA yield of 0.02 g/g under cultural conditions comprising a pH of 9.0, a temperature of 35 °C, a salinity of 3%, 2% sucrose, 0.2% peptone, and an incubation period of 48 h. The production and accumulation levels of PHA by *Bacillus* species in our study were within the range of some previous studies. For instance, Abdelmalek et al. [36] reported PHA accumulation levels of between 0.19 g/L and 2.63 g/L by *Bacillus mycoides*. However, higher production levels and concentrations of PHAs have been reported in literature for fermentation in bioreactors and production by chassis microorganisms engineered for PHA production (Table 5). For example, Mohapatra et al. [12] reported a production yield of 3.09 g/L of PHAs by *Bacillus subtilis* under optimized conditions. According to Shah and Kumar [37], yields exceeding 6.0 g/L may be produced under optimized conditions. In our study, a maximum PHA concentration of 25.9% was reported, which is lower than that achieved in well-known PHA-producing chassis microorganisms. Some studies have shown the capacity of some bacteria to accumulate PHAs at rates exceeding 80%. For instance, engineered *Halomonas* sp. SF2003 has been shown to accumulate PHAs at a rate of up to 86% from glucose [34]. *Halomonas desertis* G11 has been shown to convert 68% of biomass into PHAs [35], while *Halomonas* sp. SF2003 achieved a 78% conversion rate [38]. The high yields reported in these studies could be attributed to the use of efficient PHA-producing chassis microorganisms, production being conducted in bioreactors, and use of genetic engineering, which were absent in our study. The use of mixed microbial cultures (MCM) has also been explored as a strategy for enhancing the bacterial production of PHAs [39–41]. For instance, a PHA-accumulating MCM of *Rhodobacteraceae* bacteria has been found to have a biomass-to-PHA conversion rate of 84.1% [41]. While these strategies were not incorporated in our study, it had the major strength of offering a comprehensive evaluation of PHA-producing bacteria from halo-alkaliphilic lakes. As such, we offer useful insights into future research on the bioengineering of halo-alkaliphilic bacteria from Kenya for PHA production.

**Table 5.** Polyhydroxyalkanoates production reported in previous studies.

| Strain | Substrate | PHA Accumulation | Reference |
|---|---|---|---|
| *Halomonas bluephagenesis* | Xylose | 5.37 g/L | [42] |
| *Bacillus mycoides* | Cardboard | 2.63 g/L | [36] |
| * *Bacillus megaterium* | Glucose | 5.61 g/L | [43] |
| *Bacillus subtilis* | Sugarcane molasses | 2.5 g/L | [44] |
| * *Burkholderia cepacia* ATCC 17759 | Sugar maple | 8.72 g/L | [45] |
| *Cupriavidus necator* | Plant oils | 6.0 g/L | [46] |
| *Bacillus subtilis* | Glucose | 3.09 g/L | [12] |
| *Burkholderia sacchari* | Wastepaper | 1.6 g/L | [47] |
| ** *Bacillus* sp. strain JSM-1684023 | Sucrose | 0.516 g/L | ** |

* Fed-batch fermentation in bioreactors. ** Current study.

## 4. Materials and Methods

### 4.1. Sample Collection and Isolation of PHA-Producing Bacteria

Twenty-seven samples comprising soil, sediment, and water were collected from Lake Magadi and Lake Simbi Nyaima, located at 1.9010° S, 36.2468° E and 0.3676° S, 34.6290° E, respectively, in Kenya. For water samples, approximately 40 mL of lake water was drawn directly into 50 mL centrifuge tubes, while 300 mg samples of soil and sediment were scooped into the tubes using a sterilized hand shovel. The samples were transported to the molecular biology laboratory at the Institute for Biotechnology Research of Jomo Kenyatta University of Agriculture and Technology and refrigerated at $-20$ °C for subsequent analysis. One-gram samples of soil or sediment or 1 mL lake water samples were serially diluted and plated on minimal salt medium (MSM) [48,49] with the following components: agar 20 g/L, glucose 20 g/L, $MgSO_4$ 0.2 g/L, $Na_2HPO_4$ 4.0 g/L, $KH_2PO_4$ 2.65 g/L, $NH_4Cl$ 2.0 g/L, NaCl 30 g/L, and CaCl 0.2 g/L, and the pH of the medium was adjusted to 9.5 and 10.0 for samples from Lake Magadi and Lake Simbi, respectively. An aliquot of the dilution was spread on agar plates and incubated at 30 °C for 72 h as the primary culture. The bacterial colonies of different shapes, colors, forms, and appearances were picked from the primary culture and subcultured on fresh nutrient agar plates as the secondary culture. The colonies from the secondary culture were continuously subcultured in nutrient agar until pure colonies were obtained and maintained in 20% ($v/v$) glycerol at $-20$ °C.

### 4.2. Screening of PHA-Producing Bacteria

All isolates were subjected to a preliminary qualitative screening for PHA production using the Sudan Black B plate assay method [50,51], where forty-eight-hour-old cultures on nutrient agar were flooded with an ethanolic solution of Sudan Black B. Colonies that turned bluish black after washing with 60% ethanol were presumed to be PHA producers, while non-PHA producers were unable to retain the dye. Isolates that returned a bluish black coloration upon staining with Sudan Black B were assessed for PHA production through microscopy using the method described by Wei et al. [52], whereby the positive isolates were identified as those with black granules under the microscope. Finally, isolates that showed the presence of PHA granules under a microscope were subjected to Nile Red A staining and observed under a UV transilluminator using the Spiekermann method [53], with the PHA producers identified as those that showed orange/pinkish fluorescence.

### 4.3. Quantification of Cell Growth

Cell growth was monitored by measuring the turbidity of forty-eight-hour-old bacteria culture on a UV spectrophotometer at an optical density of 600 nm. Cell growth also involved monitoring different parameters such as carbon sources, glucose concentration, pH, and salinity to determine the optimal conditions for biomass yield.

### 4.4. PHA Production and Recovery in Submerged Fermentation

All isolates were assessed for PHA production on liquid media in 250 mL Erlenmeyer flasks. A colony of bacteria was picked from a plate and introduced into 10 mL nutrient broth that had been adjusted for pH and salinity for samples from the different lakes. The isolates were allowed to grow for 24 h in the nutrient broth to create the starter culture. For PHA accumulation, 1 mL of the starter culture was drawn and introduced in 50 mL PHA production media in 250 mL Elerlynmer flasks and allowed to grow in a rotary shaker incubator at 150 rpm at 30 °C for 72 h. The PHA production media used in this study was prepared using the following components: glucose (20 g/L), $(NH_4)_2SO_4$ (4.0 g/L), $KH_2PO_4$ (3.7 g/L), $Na_2HPO_4$ (5.8 g/L), $MgSO_4$ (0.25 g/L), and NaCl (25 g/L), and the pH was adjusted to 9.5 and 11.0 for samples from Lake Magadi and Simbi, respectively. After 72 h of growth, the cells were harvested by centrifugation at $8600\times g$ for 15 min using the method proposed by Hahn et al. [54]. The pellet was air-dried overnight to obtain a constant dry weight, which was taken and designated a dry cell weight (DCW). The dry cell pellet was suspended in a solution containing 10 mL of 10% ($v/v$) sodium hypochlorite and 10 mL

chloroform and was incubated at 37 °C for 2 h to digest the cellular components. After incubation, the suspension was centrifuged at 8600× *g* for 15 min and the bottom phase was recovered. Cellular lipids were removed by the addition of methanol and water in the ratio of 7:3 *v/v* and then centrifuged at 8500× *g* for 15 min. The supernatant was discarded, and the precipitate was washed twice with 95% (*v/v*) ethanol, air-dried overnight, and had its weight taken. The difference between the dry cell weight and the weight of the PHA pellet was designated as the residual biomass. The percentage of PHA accumulation was computed as the proportion of PHAs in a sample to the DCW.

### 4.5. Characterization of PHA-Producing Bacteria and Phylogenetic Analysis

Morphological and biochemical characterizations were performed on the bacteria that were confirmed to have the ability to produce polyhydroxyalkanoates. Morphological identification involved the observation of the morphology of bacterial colonies to delineate their structure, color, form, shape, margin, elevation, and surface characteristics. In addition, Gram reaction tests were used to differentiate between Gram-positive and Gram-negative groups. Various biochemical tests were also performed on PHA-producing bacteria, including citrate utilization, oxidase, catalase, urease activity, Voges–Proskauer, the indole production test, and the $H_2S$ test. Various carbohydrates such as starch, glucose, sucrose, xylose, mannose, cellulose, galactose, lactose, mannitol, and fructose were utilized.

For molecular characterization, genomic DNA was extracted using the Invitrogen PureLink Genomic DNA Mini Kit (Thermo Fisher Scientific, Waltham, MA, USA) following the manufacturer's instructions. The DNA was amplified using the universal primers 27F (5′ AGAGTTTGATCMTGGCTCAG 3′) and 1492 (5′TACGGYTACCTTGTTACGACTT 3′). The polymerase chain reaction was performed in a 50 μL mixture of 1 μL of the DNA template (75 ng/100 μL), 1 μL of each primer (0.2 μM), 22 μL of RNase-free water, and 25 μL of OneTaq 2X Master Mix (New England BioLabs, Ipswich, MA, USA). The PCR conditions were set as follows: initial denaturation at 95 °C for 5 min, followed by 30 cycles of 95 °C for 1 min, 55 °C for 1 min, and 72 °C for 1.5 min, and a final extension at 72 °C for 5 min. The products were resolved on 1.2% agarose gel electrophoresis and visualized on a UV transilluminator. The amplicons were purified using the QIAquick PCR purification kit (Qiagen N.V., Venlo, The Netherlands) following the manufacturer's instructions. Purified amplicons were sent to Inqaba Biotec (Pretoria, South Africa) for sequencing. Sequence consensus was obtained using the ChromasPro software. The 16S rRNA gene sequences were compared with those available on the Genbank database of the National Center for Biotechnology Information (NCBI) using the BLAST tool available at www.ncbi.nlm.nih.gov/blast (accessed on 26 April 2023) as Clustal W software was used to align the sequences with those of related bacteria strains while MEGA 7.0 was used to construct phylogenetic trees.

### 4.6. Data Analysis

Statistical analysis of PHA production was performed using Microsoft Excel 2021. The values of DCW, PHA yields, and PHA concentrations were expressed as means and standard deviations for measurements in triplicates. These means were compared. PHA concentrations was calculated as a proportion of PHA yield to DCW. PHA yields (g/g) were calculated as the proportions of PHA concentration to the amount of biomass used.

## 5. Conclusions

A diverse group of microorganisms were isolated, screened, and characterized in this study. A total of 31 potent PHA-accumulating bacteria were identified from two hypersaline and alkaliphilic lakes in Kenya: namely, Lake Simbi and Lake Magadi. Members of the genus *Bacillus* dominated the group of PHA-producing bacteria, with fewer representatives from the *Arthrobacter* spp., *Exiguobacterium* spp., *Halomonas* spp., *Paracoccus* spp., and *Rhdodobaca* spp. genera. The capacity of these bacteria to tolerate high salinity levels of up to 20% was seen as a desirable feature for minimizing contamination in fermentation procedures. A high-yielding bacteria isolate *Bacillus* sp. JSM-1684023 capable of accumulating high amounts of PHAs was illustrated in this study. Under optimized fermentation conditions, the strain produced 0.516 g/L of PHAs with a 25.9% biomass-to-PHA conversion rate, equivalent to a yield of 0.02 g/g. The optimal conditions for PHA production by the newly isolated bacteria *Bacillus* sp. JSM-1684023 were found to be an initial pH of 9.0, temperature of 35 °C, and salinity of 3%, with an incubation period of 48 h and 0.1% peptone and 2.5% sucrose as nitrogen and carbon sources, respectively. The findings of this study suggest a high potential for the production of PHAs by bacteria isolated from the hypersaline, alkaliphilic lakes of Kenya.

**Supplementary Materials:** The following supporting information can be downloaded at: https://www.mdpi.com/article/10.3390/bacteria2020007/s1, Table S1: Biochemical tests results for PHA-producing bacteria isolated from Lake Simbi; Table S2: Biochemical tests results for PHA-producing bacteria isolated from Lake Simbi.

**Author Contributions:** Conceptualization, M.N.M., S.E.A., J.M.O. and G.M.O.; data curation, M.N.M.; formal analysis, M.N.M. and G.M.O.; funding acquisition, J.M.O.; investigation, M.N.M.; methodology, M.N.M.; project administration, S.E.A. and J.M.O.; resources, S.E.A. and J.M.O.; supervision, S.E.A., J.M.O. and G.M.O.; validation, M.N.M., S.E.A., J.M.O. and G.M.O.; visualization, M.N.M.; writing—original draft, M.N.M.; writing—review and editing, S.E.A., J.M.O. and G.M.O. All authors have read and agreed to the published version of the manuscript.

**Funding:** This research was supported by funds provided by Japan International Cooperation Agency (JICA) through the Africa-ai-Japan Project, to whom we are grateful.

**Institutional Review Board Statement:** Not applicable.

**Informed Consent Statement:** Not applicable.

**Data Availability Statement:** Data collected and used to support the findings of this study are available on request from the corresponding author.

**Acknowledgments:** The authors acknowledge the Jomo Kenyatta University of Agriculture and Technology, where the research was conducted, and the Japan International Cooperation Agency (JICA) for funding the project.

**Conflicts of Interest:** The authors declare no conflict of interest. The funders had no role in the design of the study; in the collection, analyses, or interpretation of data; in the writing of the manuscript; or in the decision to publish the results.

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
