# Peer review of "The Isolation, Screening, and Characterization of Polyhydroxyalkanoate-Producing Bacteria from Hypersaline Lakes in Kenya"

_2674-1334, doi:10.3390/bacteria2020007_

Round 1

Reviewer 1 Report

The manuscript by Muigano and co-authors present the isolation and characterization of novel PHA producing bacteria from saline environments. The study is of high significance, but some issues should be solved prior to publication.

Specific comments

L 20-22. Please specify the carbon source in this case.

L 43. A more adequate reference for this statement is an original and classic manuscript - doi: 10.1038/191463a0

L 47-51. There is extensive research on the use of haliphilic strains for PHA production, and it should be present in the introduction and discussion section. Please consider doi.org/10.1002/adbi.201800088, doi.org/10.1016/j.copbio.2017.11.016 for example.

L 55. Does the 37 PHA-positive samples were isolated from the same environment? Do you have more information that could point to a condition in which more PHA+ organisms were observed?

L 66. Figure quality is not ideal. The pictures should be taken with a white clean background.

L 69. Please consider pointing to the granules. It is not obvious in this figure.

L 80. 13% of cell dry weight? Please clarify why this value was the threshold.

L 81-82. Color in which media?

L 82-83. Please include the results for the 31 analyzed samples as Supplementary material.

L 87. Is it really a fermentative process? Please review. Same for line 93. Please consider using catabolize in the entire text.

L 88-89. Include all the obtained results as supplementary material.

L 90-91. It is not clear why table 1 shows only 4 isolates, is there a reason for that?

L 91. Identified by which technique? Same for L94.

L 114. Table 2 - please use italics for all scientific nomenclature. Review the entire text and reference list.

L 119. Please use italics for all scientific nomenclature. Review the entire text and reference list.

L 127. Consider using different colors for samples from each lake. This would help the reader.

L 135-138. It would be more adequate to present the PHA yield from carbon source (in g/g) on a separate table. Please check doi.org/10.1007/s002530050763 for such a table.

L 138. Compare the obtained values with the maximum theoretical yield from each used carbon source. Present how much of the maximum productivity was achieved in the current conditions.

L 149. How was this identified? Preliminary experiments?

L 151-152. Not clear. According to the Figure 6, MO22 presented better growth on glucose, sucrose and mannitol.

L 159. Not clear why glucose was selected.

L 165. Figure 7 is Figure 6. Please correct for adequate peer-review.

L 173. At which cultivation time point was the OD measured? Please clarify.

L 178. Reference missing for Halomonas alkalikola growth data. Clarify if this data was generated in this work.

L 182. At which cultivation time point was the OD measured? Please clarify.

L 213-214. Compare the obtained data with classical PHA producing chassis, such as Halomonas bluephagenesis, Bacillus subtilis (10.1186/1475-2859-8-38), Burkholderia sacchari, etc. Consider presented this in a comparative Table also.

L 221-223. Add conversion yields and compare to the maximum theoretical value and with other strains (literature).

L 246. The original Sudan Black B research manuscript must be cited doi.org/10.1007/BF00410161

L 269-271. Please add a reference for the media composition if it is not an original media.

L 300. Please add primer and DNA concentrations.

Reviewer 2 Report

My comments are added in attached document. Nice study but revisions are needed

Reviewer 3 Report

Muigano et al isolated, screened, and characterized polyhydroxyalkanoates-producing bacteria from hypersaline lakes in Kenya. They found that two isolates have the highest PHA production capacities with concentrations of 21 16.14% and 19.14%, respectively. Halomonas alkalikola and Bacillus sp. JSM-1684023 yielded 0.397 g/L 22 and 0.29 g/L of PHAs, respectively on a 2% glucose substrate. Overall, the manuscript was well organized and their findings are desirable for researchers around the world. However, there are some minor concerns about this study that warrant consideration prior to publication. Please see my comments below.

1.    For Table 1, I would suggest to replace the lines between the line under “Colony shape” to the line above “Mannitol” by using shade every other rows, which may make the Table 1 less busier.

2.    For Figure 2, it is better to add a measurement there to show the size of the image.

3.    For Figures 5-7, the frames of the figures could be removed.

4.    For Figure 5, it may help visualize if using colorful columns and lines.

5.    For Figures 8 and 9, they looked blurry, I would suggest to re-make them.  

6.    For Line 236, the numbers in both KH2PO4 and NH4Cl are supposed to use subscripts.

7.    For Lines 149-150, I would suggest to add a couple of references to support the claim.

Round 2

Reviewer 1 Report

The authors replied to almost all the comments.

I invite the authors to revise the definition of yield, since there is a conceptual mistake. Yield refers to grams of product produced per 1 gram of substrate, i.e. you can calculate how many grams of 3HB monomer are produced from 1 g of carbon source (glucose, in your case). Titer refers to the produced PHA in g/L. You can find many examples of titer and yield data recently published on different journals: https://doi.org/10.1016/j.nbt.2016.10.001, doi: 10.3390/bioengineering4020036

The maximum theoretical yield can be calculated according to your carbon source- you can check the reference 'The Handbook of Polyhydroxyalkanoates' for more information - doi.org/10.1201/9780429296635

This correction will increase manuscript's quality and enhance its visibility in the PHA community.

Author Response

I revised the definition of yield throughout the paper. What I had initially referred to as PHA Yield was in fact PHA Concentration and I made the correction in all parts of the paper and the charts. I also included calculations for PHA Yield and compared our results with the theoretical yields. The resources provided were extremely useful in correcting the conceptual mistake that was present in the earlier draft and I am grateful for the insights provided by the reviewer.

Reviewer 2 Report

Article has much improved. Comments are properly addressed

Author Response

I reviewed the entire paper and noted some minor issues on grammar, spelling errors, and sentence structure. I corrected all the issues and I believe the paper is now adequately improved.